# Sparc3D: Sparse Representation and Construction for High-Resolution 3D Shapes Modeling

**Zhihao Li**[1,2*], **Yufei Wang**[1], **Heliang Zheng**[2,‡], **Yihao Luo**[2,3,†], **Bihan Wen**[1,†]

[1]Department of EEE, Nanyang Technological University, Singapore
[2]Math Magic   [3]Imperial-X, Imperial College London, UK

zhihao005@e.ntu.edu.sg, yufei001@e.ntu.edu.sg, zhenghllj@gmail.com
y.luo23@imperial.ac.uk, bihan.wen@ntu.edu.sg

https://lizhihao6.github.io/Sparc3D

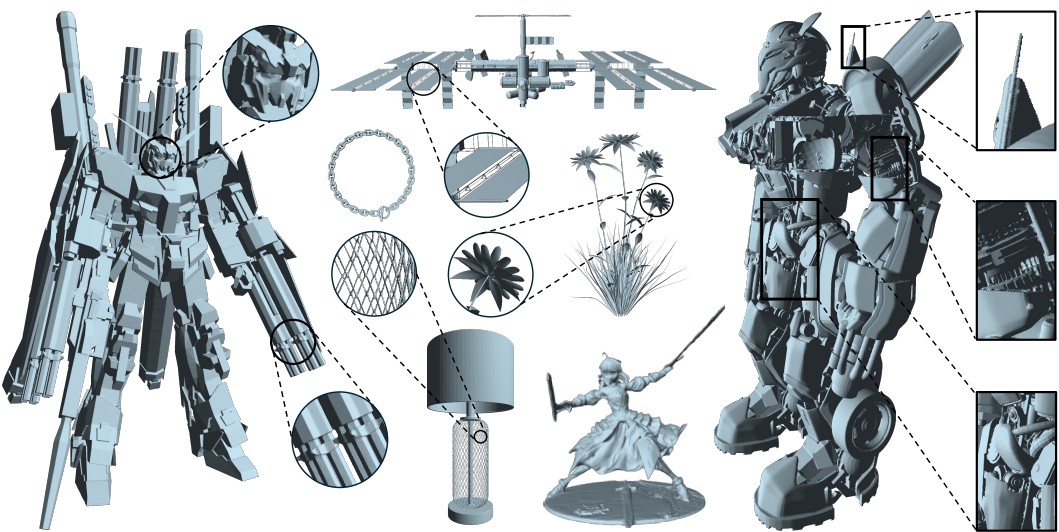

Figure 1: **Sparc3D Reconstruction Results.** Leveraging our sparse deformable marching cubes (**Sparcubes**) representation and sparse convolutional VAE (**Sparconv-VAE**), our method achieves state-of-the-art reconstruction quality on challenging 3D inputs. It robustly handles open surfaces (automatically closed into watertight meshes), recovers hidden interior structures, and faithfully reconstructs highly complex geometries (see zoom-in views, top to bottom). All outputs are fully watertight and 3D-printable, demonstrating the potential of our framework for high-resolution 3D mesh generation. *Best viewed with zoom-in.*

## Abstract

High-fidelity 3D object synthesis remains significantly more challenging than 2D image generation due to the unstructured nature of mesh data and the cubic complexity of dense volumetric grids. Existing two-stage pipelines—compressing meshes with a VAE (using either 2D or 3D supervision), followed by latent diffusion sampling—often suffer from severe detail loss caused by inefficient representations and modality mismatches introduced in VAE. We introduce **Sparc3D**, a unified framework that combines a sparse deformable marching cubes representation **Sparcubes**

---

*This work was conducted during Zhihao Li's research internship at Math Magic.
†Corresponding authors; ‡:project lead.

with a novel encoder **Sparconv-VAE**. Sparcubes converts raw meshes into high-resolution ($1024^3$) surfaces with arbitrary topology by scattering signed distance and deformation fields onto a sparse cube, allowing differentiable optimization. Sparconv-VAE is the first modality-consistent variational autoencoder built entirely upon sparse convolutional networks, enabling efficient and near-lossless 3D reconstruction suitable for high-resolution generative modeling through latent diffusion. Sparc3D achieves state-of-the-art reconstruction fidelity on challenging inputs, including open surfaces, disconnected components, and intricate geometry. It preserves fine-grained shape details, reduces training and inference cost, and integrates naturally with latent diffusion models for scalable, high-resolution 3D generation.

# 1   Introduction

Recent breakthroughs in 3D object generation [2, 15, 17, 24, 29, 32, 39] have enabled applications in virtual domains, such as AR/VR [3, 14, 16, 20] and robotics simulation [34, 35]—as well as in physical contexts like 3D printing [29]. Despite this progress, synthesizing high-fidelity 3D assets remains far more challenging than generating 2D imagery or text stemming from the inherently unstructured nature of 3D data and the cubic scaling of dense volumetric representations.

Drawing on the success of text-to-image diffusion models [25], many 3D generation pipelines [2, 15, 17, 32, 39] employ a two-stage process: a variational autoencoder (VAE) followed by latent diffusion. Most current VAEs employ either 3D supervision[2, 15, 17, 39]—typically via a global vector set representation—or 2D supervision[32]—commonly via a sparse-voxel representation—but both suffer from limited resolution and a modality mismatch between inputs and outputs.

3D supervised VAEs [2, 15, 17, 39] require watertight meshes for sampling supervision signals, yet most raw meshes are not watertight and must be remeshed. The common pipeline [2, 15, 39] shown in Fig. 2, first samples an Unsigned Distance Function (UDF) on a voxel grid, then approximates a Signed Distance Function (SDF) by subtracting two voxel sizes—halving the effective resolution and introducing errors that propagate through both VAE reconstruction and diffusion generation. After applying Marching Cubes [19] or Dual Marching Cubes [26], the mesh becomes double-layered, necessitating an additional step to retain only the largest connected component. This step inadvertently discards smaller yet crucial features, misaligning the conditioned input image from raw mesh with the reconstructed 3D shape during diffusion training.

Most recently, TRELLIS [32] demonstrated the possibility of training a 3D VAE using only 2D supervision. While it avoids degradation from mesh conversion, it still depends on dense volumetric grids for 2D projections, which limit resolution. Moreover, lacking any 3D topological constraints, the generated object's interior geometry may be incorrect and its surfaces can remain open—a critical flaw for applications such as 3D printing.

All of these VAEs also contend with a fundamental modality gap: 3D supervised methods [2, 15, 17, 39] ingest surface points and normals but decode SDF values, while TRELLIS [32] encodes voxelized DINOv2 [23] features into SDF. Bridging this gap requires heavy attention mechanisms, which increase model complexity and risk of amplifying underlying inconsistencies.

In this work, we introduce **Sparcubes** (Sparse Deformable Marching Cubes), a fast, near-lossless pipeline for converting raw meshes into watertight surfaces. Our method begins by identifying a sparse set of activated voxels from the input mesh and performing a flood-fill to assign coarse signed labels. We then optimize grid-vertex deformations via gradient descent and refine them using a view-dependent 2D rendering loss. Sparcubes converts a raw mesh into a watertight $1024^3$ grid in under 30 seconds—achieving a threefold speedup over prior watertight conversion methods [2, 15] without sacrificing fine details or small components.

Building on Sparcubes, we introduce **Sparconv-VAE**, a modality-consistent VAE composed of a sparse encoder and a self-pruning decoder. By eliminating the modality gap, Sparconv-VAE can employ a lightweight architecture without relying on heavy global attention mechanisms. Experimental results show that our VAE achieves state-of-the-art reconstruction performance and minimal training cost. Furthermore, it seamlessly integrates with existing latent diffusion pipelines such as TRELLIS [32], further enhancing the resolution of the generated 3D objects.

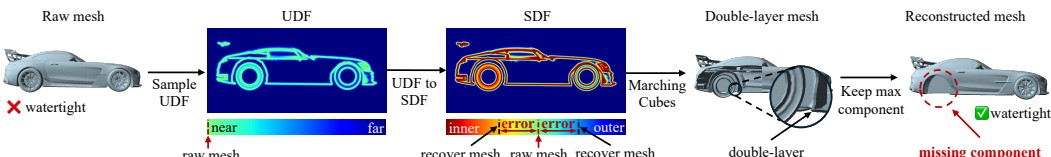

Figure 2: **Problems of the previous SDFs extraction pipeline.** The widely used SDFs extraction workflow [2, 15, 39] suffers from two critical failures: resolution degradation (show as error) and missing geometry (circled on the right). Converting UDF to SDF by subtracting two voxel sizes effectively halves the spatial resolution. Moreover, the SDF extraction yields a double-layer mesh, from which only the largest connected component is retained, inadvertently discarding smaller but important component. Together, these two deficiencies substantially limit the upper-bound performance of downstream VAEs and generation models. *Best viewed with zoom-in.*

Our main contributions can be summarized as:

- We propose **Sparcubes**, a fast, near-lossless remeshing algorithm that converts raw meshes into watertight surfaces at $1024^3$ resolution in approximately $30$ s, achieving a $3\times$ speedup over prior methods without sacrificing any components.

- We introduce **Sparconv-VAE**, a modality-consistent variational autoencoder that employs a sparse convolutional encoder and a self-pruning decoder. By eliminating the input–output modality gap, our architecture achieves high computational efficiency and near-lossless reconstruction, without global attention.

- Experimental results show that, our **Sparc3D** framework, comprising Sparcubes and Sparconv-VAE, achieves state-of-the-art reconstruction fidelity, reduces training cost, and seamlessly integrates with current latent diffusion frameworks to enhance the resolution of generated 3D objects.

## 2 Related Work

### 2.1 3D Shape Representation and Generation

**Mesh and point cloud.**    Triangle meshes and point clouds are the most common representations of 3D data. Triangle meshes, composed of vertices and triangular faces, offer precise modeling of surface detail and arbitrary topology. However, their irregular graph structure complicates learning, requiring neural networks to handle non-uniform neighborhoods, varying vertex counts, and the absence of a canonical ordering. To address this, recent methods [4, 5, 11, 28, 30] adopt autoregressive models to jointly generate geometry and connectivity, though these suffer from limited context length and slow sampling. In contrast, point clouds represent surfaces as unordered sets of 3D points, making them easy to sample from distributions [21, 31, 33], but difficult to convert directly into watertight surfaces due to the lack of explicit connectivity [9, 36].

**Isosurface.** Isosurface is a continuous surface representing a mesh boundary via a signed distance field (SDF). Most methods [13, 19, 26, 27] subdivide space into voxels, polygonize each cell, then stitch them into a mesh. Marching Cubes uses a fixed lookup table but can suffer topological ambiguities [19]. Dual Marching Cubes (DMC) fixes this by placing vertices on edges where the isosurface crosses and linking them via Dual Contouring, yielding watertight meshes [13, 26]. Both rely on a uniform cube size, limiting detail; FlexiCubes [27] deforms the grid and applies isosurface slope weights to adapt voxel sizes to local geometry, improving accuracy.

Many 3D generation methods adopt SDF-based supervision [2, 15, 17, 24, 32, 38, 39]. Techniques relying solely on 2D supervision [32] may implicitly learn an SDF, but often produce open surfaces or incorrect interiors due to the lack of volumetric constraints. In contrast, fully 3D-supervised approaches [2, 15, 17, 24, 38, 39] extract explicit SDFs from meshes with arbitrary topology, making accurate and adaptive SDF extraction a critical but challenging component for high-fidelity reconstruction.

## 2.2 3D Shape VAEs

**VecSet-based VAEs.** VecSet-based methods represent 3D shapes as sets of global latent vectors constructed from local surface features. 3DShape2VecSet [37] embeds sampled points and normals into a VecSet using a transformer and supervises decoding via surrounding SDF values. CLAY [38] scales the architecture to larger datasets and model sizes, while TripoSG [18] enhances expressiveness through mixture-of-experts (MoE) modules. Dora [2] and Hunyuan2 [39] improve sampling by prioritizing high-curvature regions. Despite these advances, all approaches face a modality mismatch: local point features are compressed into global latent vectors and decoded back to local fields, forcing the VAE to perform both feature abstraction and modality conversion, which increases reliance on attention and model complexity.

**Sparse voxel–based VAEs.** In contrast, sparse voxel-based VAEs preserve spatial structure by converting meshes into sparse voxel grids with feature vectors. XCube [24] replaces the global VecSet in 3DShape2VecSet [37] with voxel-aligned SDF and normal features, improving detail preservation. TRELLIS [32] enriches this representation with aggregated DINOv2 [23] features, enabling joint modeling of 3D geometry and texture. TripoSF [12] further scales the framework to high-resolution reconstructions (see Supplementary Material for details). Nonetheless, these methods still face the challenge of modality conversion—mapping point normals or DINOv2 descriptors to continuous SDF fields remains a key bottleneck.

# 3 Method

## 3.1 Preliminaries

**Distance fields.** A distance field is a scalar function $\Phi : \mathbb{R}^3 \to \mathbb{R}$ that measures the distance to a surface. The unsigned distance function (UDF) encodes only magnitude, while the signed distance function (SDF) adds sign to distinguish interior and exterior:

$$\text{UDF}(\mathbf{x}, \mathcal{M}) = \min_{\mathbf{y} \in \mathcal{M}} \|\mathbf{x} - \mathbf{y}\|_2, \quad \text{SDF}(\mathbf{x}, \mathcal{M}) = \text{sign}(\mathbf{x}, \mathcal{M}) \cdot \text{UDF}(\mathbf{x}, \mathcal{M}), \tag{1}$$

where $\text{sign}(\mathbf{x}, \mathcal{M}) \in \{-1, +1\}$ indicates inside/outside status. For non-watertight or non-manifold meshes, computing $\text{sign}$ is non-trivial [1].

**Marching cubes and sparse variants.** The marching cubes algorithm [19] extracts an isosurface mesh from a volumetric field $\Phi$ by interpolating surface positions across a voxel grid:

$$\mathcal{S} = \{\mathbf{x} \in \mathbb{R}^3 \mid \Phi(\mathbf{x}) = 0\}. \tag{2}$$

Sparse variants [10] operate on narrow bands $|\Phi(\mathbf{x})| < \epsilon$ to reduce memory usage. We further introduce a sparse cube grid $(V, C)$, where $V$ is a set of sampled vertices and $C$ contains 8-vertex cubes. Deformable and weighted dual variants, exemplified by FlexiCubes [27], extend this process by modeling the surface as a deformed version of the sparse cube grid. Specifically, each grid node $n_i$ in the initial grid is displaced to a new position $n_i + \Delta n_i$, forming a refined grid $(N + \Delta N, C, \Phi, W)$ that better conforms to the implicit surface, where the displacements $\Delta n_i$ and per-node weights $w_i \in W$ are learnable during optimization.

## 3.2 Sparcubes

Our method, **Sparcubes** (Sparse Deformable Marching Cubes), reconstructs watertight and geometrically accurate surfaces from arbitrary input meshes through sparse volumetric sampling, coarse-to-fine sign estimation, and deformable refinement. Unlike dense voxel methods, Sparcubes represents geometry using a sparse set of voxel cubes, where each cube vertex carries a signed distance value. This representation enables efficient computation, memory scalability, and supports downstream surface extraction or direct use in learning-based pipelines.

As shown in Fig. 3, the core pipeline consists of the following steps:

**Step 1: Active voxel extraction and UDF computation.** We begin by identifying a sparse set of *active voxels* within a narrow band around the input surface. These are voxels whose corner vertices

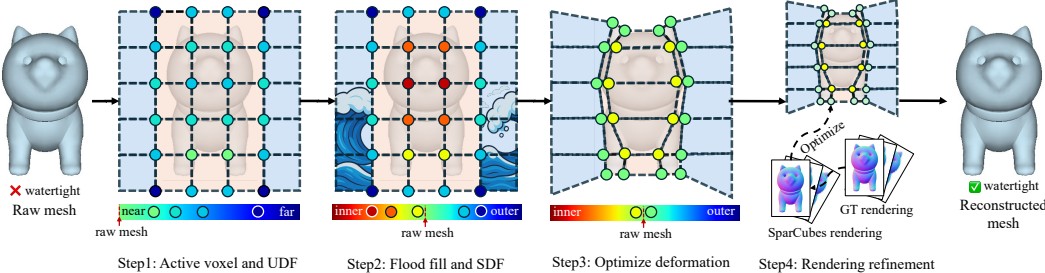

Figure 3: Illustration of our Sparcubes reconstruction pipeline for converting a raw mesh into a watertight mesh.

lie within a threshold distance $\epsilon$ from the mesh $\mathcal{M}$. For each corner vertex $\mathbf{x} \in \mathbb{R}^3$, we compute the unsigned distance to the surface:

$$\text{UDF}(\mathbf{x}) = \min_{\mathbf{y} \in \mathcal{M}} \|\mathbf{x} - \mathbf{y}\|_2. \tag{3}$$

This yields a sparse volumetric grid $\Phi$, with distance values concentrated near the surface geometry, suitable for efficient storage and processing.

**Step 2: Flood fill for coarse sign labeling.** To convert the unsigned field into a signed distance function (SDF), we apply a volumetric flood fill algorithm [22] starting from known exterior regions (*e.g.*, corners of the bounding box). This produces a binary occupancy label $T(\mathbf{x}) \in \{0, 1\}$, indicating whether point $\mathbf{x}$ is inside or outside the shape. We then construct the coarse signed distance field as:

$$\text{SDF}(\mathbf{x}) = (1 - 2T(\mathbf{x})) \cdot \text{UDF}(\mathbf{x}), \tag{4}$$

which gives a consistent sign assignment under simple labeling and forms the basis for further refinement.

**Step 3: Gradient-based deformation optimization.** Instead of explicitly refining a globally accurate SDF, we directly optimize the geometry of the sparse cube structure to better conform to the underlying surface. Given an initial volumetric representation $(V, C, \Phi_v)$—where $V$ denotes the set of sparse cube corner vertices, $C$ is the set of active cubes, and $\Phi$ is the signed distance field defined at each vertex—we perform a geometric deformation to obtain $(V + \Delta V, C, \Phi_v)$. This results in a geometry-aware sparse SDF volume that more accurately approximates the zero level set of the implicit surface. Notably, for points where $\Phi(x) > 0$, the SDF values are often only coarse approximations, particularly in regions far from the observed surface or near topological ambiguities. These regions may exhibit significant errors due to poor connectivity, occlusions, or non-watertight input geometry. As such, rather than refining $\Phi$ globally, we optimize the vertex positions $\Delta V$ to implicitly correct the spatial alignment of the zero level set. To improve the accuracy of sign estimation and geometric alignment, we displace each vertex slightly along the unsigned distance field gradient:

$$\mathbf{x}' = \mathbf{x} - \eta \cdot \nabla \text{UDF}(\mathbf{x}). \tag{5}$$

This heuristic captures local curvature and topological cues that are not easily recovered through purely topological methods such as flood fill. It also allows us to estimate sign information in regions with ambiguous connectivity, such as thin shells or open surfaces. The final data structure is a sparse cube grid with SDF values on each corner $(V, C, \Phi_v, \Delta V)$, denoted as **Sparcubes**.

**Step 4: Rendering-based refinement.** Sparcubes supports differentiable mesh extraction, enabling further end-to-end refinement with perceptual signals. When multi-view images, silhouettes, or depth maps are available, we optionally introduce a differentiable rendering loss to further enhance visual realism and geometric alignment. Given a reconstructed mesh $\mathcal{M}_r$ extracted from the deformed Sparcubes, we compute a multi-term rendering loss:

$$\mathcal{L}_{\text{render}} = \|\mathcal{R}^D(\mathcal{M}_r) - \mathcal{I}^D_{\text{obs}}\|_2^2 + \|\mathcal{R}^N(\mathcal{M}_r) - \mathcal{I}^N_{\text{obs}}\|_2^2, \tag{6}$$

where $\mathcal{R}^D(\mathcal{M}_r)$ denotes the rendered depth image of the mesh under known camera parameters, and $\mathcal{R}^N(\mathcal{M}_r)$ is the corresponding rendered normal map. The terms $\mathcal{I}^D_{\text{obs}}$ and $\mathcal{I}^N_{\text{obs}}$ are the observed depth image and the ground truth normal map derived from the input or canonical mesh. Leveraging our voxel-based data structure, we can easily identify visible voxels and render exclusively within those regions, greatly reducing computational cost.

### 3.3 Sparconv-VAE

Building on our Sparcubes representation, we develop **Sparconv-VAE**, a sparse convolution-based variational autoencoder without high-consuming global attentions, which directly compresses the Sparcubes parameters $\{\phi \in \Phi_v, \delta \in \Delta V\}$ into a sparse latent feature $\mathbf{z}$ and decodes back to the same format without any modality conversion.

**Architecture and Loss Function.** Our encoder is a cascade of sparse residual convolutional blocks that progressively downsample the input features. At the coarsest resolution, a lightweight local attention module aggregates neighborhood information. The decoder mirrors this process, interleaving sparse residual convolutions with self-pruning upsample blocks to restore the original resolution and predict the Sparcubes parameters $\{\hat{\phi}, \hat{\delta}\}$. Each self-pruning block first predicts the occupancy of the subdivided voxel occupancy mask $\mathbf{o}$, supervised by $\mathcal{L}_{\mathrm{occ}} = \mathrm{BCE}(\hat{\mathbf{o}}, \mathbf{o})$, then applies a learned upsampling to refine the voxel features. Because $\phi$ is sign-sensitive (inside vs. outside), we split its prediction into a sign branch and a magnitude branch. The sign branch predicts $\mathrm{sign}(\hat{\phi})$ under $\mathcal{L}_{\phi_{\mathrm{sign}}} = \mathrm{BCE}(\mathrm{sign}(\hat{\phi}), \mathrm{sign}(\phi))$, while the magnitude branch regresses $\hat{\phi}$ with $\mathcal{L}_{\phi_{\mathrm{mag}}} = \|\hat{\phi}, \phi\|_2$. The deformation vectors are optimized via $\mathcal{L}_\delta = \|\hat{\delta}, \delta\|_2$. Finally, we regularize the latent distribution using the VAE's Kullback–Leibler divergence $\mathcal{L}_{\mathrm{KL}} = \mathrm{KL}(q(\mathbf{z}|\delta, \phi)\|p(\mathbf{z}))$, yielding a single cohesive training objective that jointly minimizes the occupancy, sign, magnitude, deformation and KL divergence losses:

$$\mathcal{L} = \lambda_{\mathrm{occ}} \mathcal{L}_{\mathrm{occ}} + \lambda_{\mathrm{sign}} \mathcal{L}_{\phi_{\mathrm{sign}}} + \lambda_{\mathrm{mag}} \mathcal{L}_{\phi_{\mathrm{mag}}} + \lambda_\delta \mathcal{L}_\delta + \lambda_{\mathrm{KL}} \mathcal{L}_{\mathrm{KL}}. \tag{7}$$

A detailed description of the module design and the choice of $\lambda$ are in the Supplementary Material.

**Hole filling.** Although predicted occupancy may be imperfect and introduce small holes, our inherently watertight Sparcubes representation allows for straightforward hole detection and filling. We first identify boundary half-edges. For each face $\mathbf{f} = \{\mathbf{v}_0, \mathbf{v}_1, \mathbf{v}_2\}$, we emit the directed edges $(\mathbf{v}_0 \rightarrow \mathbf{v}_1), (\mathbf{v}_1 \rightarrow \mathbf{v}_2)$, and $(\mathbf{v}_2 \rightarrow \mathbf{v}_0)$. By sorting each pair of vertices into undirected edges and counting occurrences, edges whose undirected counterpart appears only once are marked as boundary edges. We build an outgoing-edge map keyed by source vertex, and then recover closed boundary loops by walking each edge until returning to its start. To triangulate each boundary loop $\mathcal{C} = \{\mathbf{v}_i\}_{i=1}^n$, we follow a classic ear-filling pipeline: compute a geometric score at every vertex, fill the "best" ear, and repeat until all open small boundaries vanish. Specifically, the score on each pending filled angle $A_i$ is defined by

$$A_i = \mathrm{atan2}(\|\mathbf{d}_{i-1 \rightarrow i} \times \mathbf{d}_{i \rightarrow i+1}\|_2, -\mathbf{d}_{i-1 \rightarrow i} \cdot \mathbf{d}_{i \rightarrow i+1}). \tag{8}$$

In each iteration, we select the vertex with the smallest $A_i$ (*i.e.*, the sharpest convex ear), form the triangle $(\mathbf{v}_{i-1}, \mathbf{v}_i, \mathbf{v}_{i+1})$, and update the boundary. Merging all new triangles with the original face set closes every small hole.

## 4 Experiments

### 4.1 Experiment Settings

**Implementation details.** We implement all Sparcubes as custom CUDA kernels. Following TREL-LIS [32], we train both the Sparconv-VAE and its latent flow model on 500 K high-quality assets from Objaverse [8] and Objaverse-XL [7]. The VAE runs on 32 A100 GPUs (batch size 32) with AdamW (initial LR $1 \times 10^{-4}$) for two days. We then fine-tune the TRELLIS latent flow model on our VAE latents using 64 A100 GPUs (batch size 64) for ten days. At inference, we sample with a classifier-free guidance scale of 3.5 over 25 steps, matching TRELLIS settings.

**Dataset.** Following Dora [2], we curated a VAE test set by selecting the most challenging examples from the ABO [6] and Objaverse [8] datasets—specifically those exhibiting occluded components, intricate geometric details, and open surfaces. To avoid any overlap with training data used in prior work, we additionally assembled a "Wild" dataset with multiple components from online sources that is disjoint from both ABO and Objaverse. For generation, we also benchmarked our method against TRELLIS [32] using the wild dataset.

**Compared methods.** We compare our Sparconv-VAE with the previous sate-of-the art vaes, including TRELLIS [32], Craftsman [15], Dora [2] and XCubes [24]. Because our diffusion architecture and

Table 1: **Quantitative comparison of watertight remeshing across the ABO [6], Objaverse [8], and In-the-Wild datasets.** Chamfer Distance (CD, $\times 10^4$), Absolute Normal Consistency (ANC, $\times 10^2$) and F1 score (F1, $\times 10^2$) are reported.

| Method | ABO [6] | | | Objaverse [8] | | | Wild | | |
|---|---|---|---|---|---|---|---|---|---|
| | CD ↓ | ANC ↑ | F1 ↑ | CD ↓ | ANC ↑ | F1 ↑ | CD ↓ | ANC ↑ | F1 ↑ |
| Dora-wt-512 [2] | 1.16 | 76.94 | 83.18 | 4.25 | 75.77 | 61.35 | 67.2 | 78.51 | 64.99 |
| Dora-wt-1024 [2] | 1.07 | 76.94 | 84.56 | 4.35 | 75.04 | 63.84 | 63.7 | 78.77 | 65.90 |
| Ours-wt-512 | 1.01 | 77.75 | 85.21 | 3.09 | 75.35 | 64.81 | 0.47 | 88.58 | 96.95 |
| Ours-wt-1024 | 1.00 | 77.66 | 85.39 | 3.01 | 74.98 | 65.65 | 0.46 | 88.55 | 97.06 |

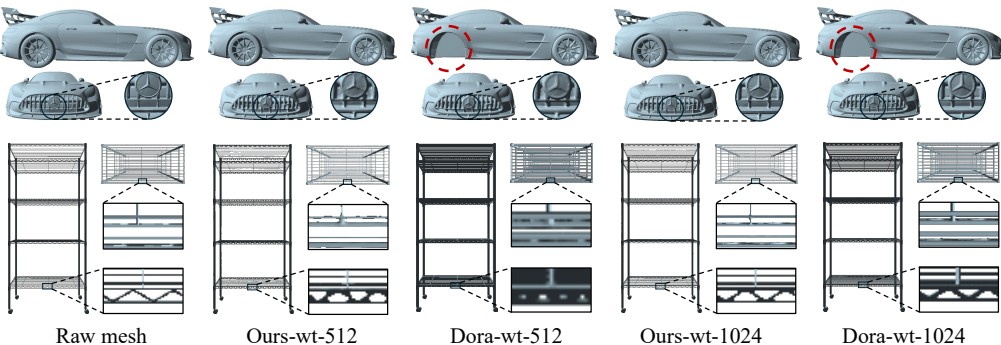

Raw mesh     Ours-wt-512     Dora-wt-512     Ours-wt-1024     Dora-wt-1024

Figure 4: **Qualitative comparison of watertight remeshing pipelines.** We evaluate our Sparcubes remeshing pipeline against previous widely used one [2, 15, 39], *i.e.*, Dora-wt [2], at voxel resolutions of 512 and 1024. Compared with the previous method, our Sparcubes preserves crucial components (e.g., the car wheel) and recovers finer geometric details (e.g., the shelving frame). Our wt-512 result even outperforms the wt-1024 remeshed by Dora-wt [2]. *Best viewed with zoom-in.*

model size match those of TRELLIS [32], we evaluate our generation results against it to ensure a fair comparison.

## 4.2 Comparation Results

**Watertight remeshing results.** We evaluate both our watertight remeshing (serving as VAE ground truth) on the ABO [6], Objaverse [8], and Wild datasets, using Chamfer distance (CD), Absolute Normal Consistency (ANC), and F1 score (F1) as metrics. As Table 1 shows, our Sparcubes consistently outperforms prior pipelines [2, 15, 39] (reported under "Dora-wt" [2] for brevity) across all datasets and metrics. Remarkably, our wt-512 remeshed outputs even exceed the quality of the wt-1024 results produced by previous methods. Fig. 4 presents qualitative comparisons: our approach faithfully preserves critical components (*e.g.*, the car wheel) and recovers fine geometric details (*e.g.*, shelving frames).

**VAE reconstruction results.** We further assess our Sparconv-VAE reconstruction against TREL-LIS [32], Craftsman [15], Dora [2], and XCubes [24] in Table 2. Across the majority of datasets and metrics, our Sparconv-VAE outperforms these prior methods. Qualitative results in Fig. 5 illustrate that our VAE faithfully reconstructs complex shapes with fine details (all columns), converts open surfaces into double-layered watertight meshes (columns 1, 4, and 6), and reveals unvisible hidden internal structures (column 6).

**Generation results.** We also valid the effectiveness of our Sparconv-VAE for generation by fine-tuning the TRELLIS [32] pretrained model. Under the same diffusion architecture and model size (see Fig. 6), our approach synthesizes watertight 3D shapes with exceptional fidelity and rich detail—capturing, for example, the sharp ridges of pavilion eaves, the subtle facial features of human figures, and the intricate structural elements of robots.

Table 2: **Quantitative comparison of VAE reconstruction across the ABO [6], Objaverse [8], and In-the-Wild datasets.** Chamfer Distance (CD, $\times 10^4$), Absolute Normal Consistency (ANC, $\times 10^2$) and F1 score (F1, $\times 10^2$) are reported.

| Method | ABO [6] | | | Objaverse [8] | | | Wild | | |
| --- | --- | --- | --- | --- | --- | --- | --- | --- | --- |
| | CD $\downarrow$ | ANC $\uparrow$ | F1 $\uparrow$ | CD $\downarrow$ | ANC $\uparrow$ | F1 $\uparrow$ | CD $\downarrow$ | ANC $\uparrow$ | F1 $\uparrow$ |
| TRELLIS [32] | 1.32 | 75.48 | 80.59 | 4.29 | 74.34 | 59.27 | 0.70 | 85.60 | 94.04 |
| Craftsman [15] | 1.51 | 77.46 | 77.47 | 2.53 | 77.37 | 55.28 | 0.89 | 87.81 | 92.28 |
| Dora [2] | 1.45 | 77.21 | 78.54 | 4.85 | 77.19 | 54.37 | 68.2 | 78.79 | 62.07 |
| XCubes [24] | 1.42 | 65.45 | 77.57 | 3.67 | 61.81 | 51.65 | 2.02 | 62.21 | 73.74 |
| Ours-512 | 1.01 | 78.09 | 85.33 | 3.09 | 75.59 | 64.92 | 0.47 | 88.74 | 96.97 |
| Ours-1024 | 1.00 | 77.69 | 85.41 | 3.00 | 75.10 | 65.75 | 0.46 | 88.70 | 97.12 |

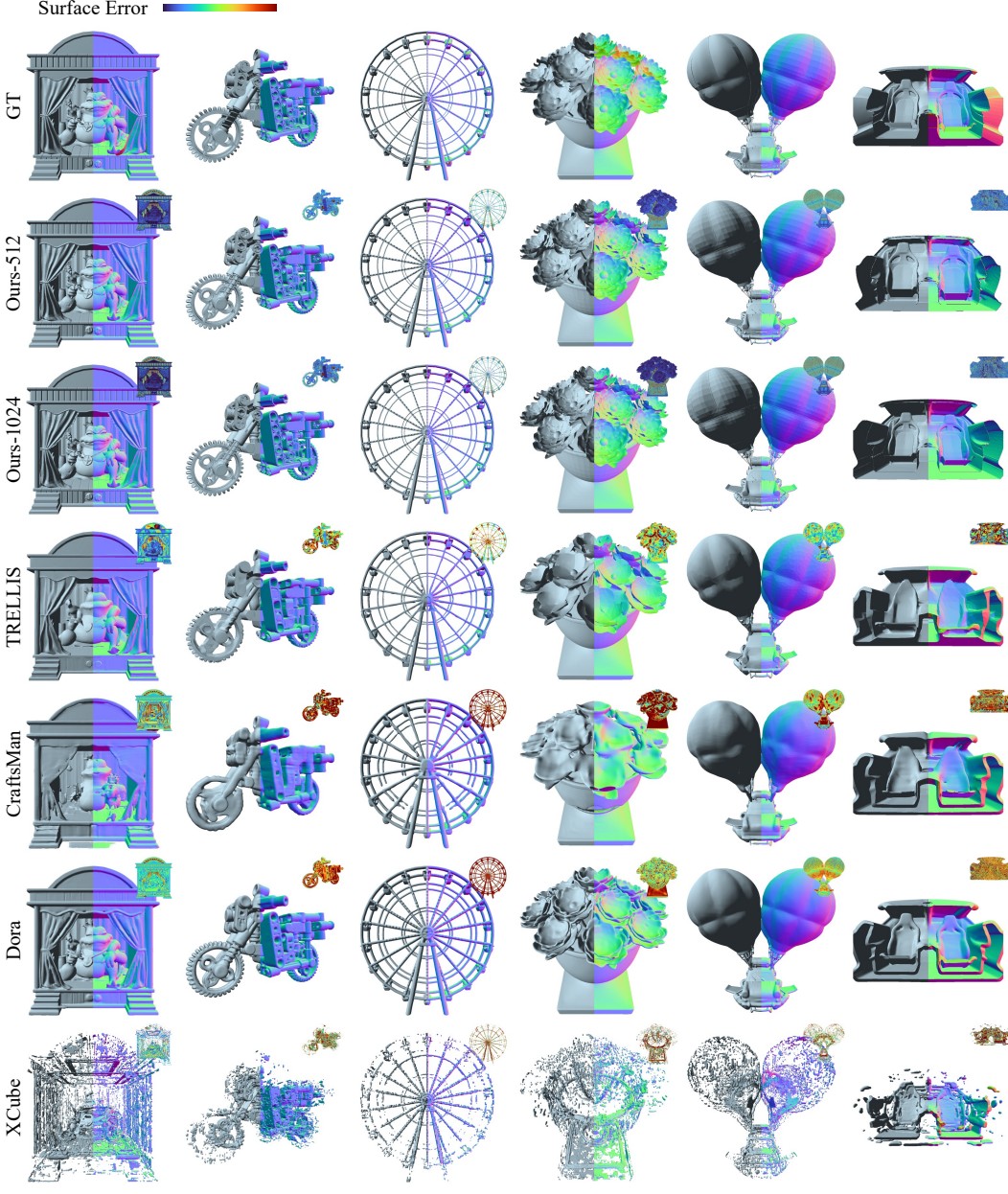

Figure 5: **Qualitative comparison of VAE reconstructions.** Our Sparconv-VAE demonstrates superior performance in reconstructing complex geometries, converting open surfaces into double-layered watertight meshes, and revealing unvisible internal structures. *Best viewed with zoom-in.*

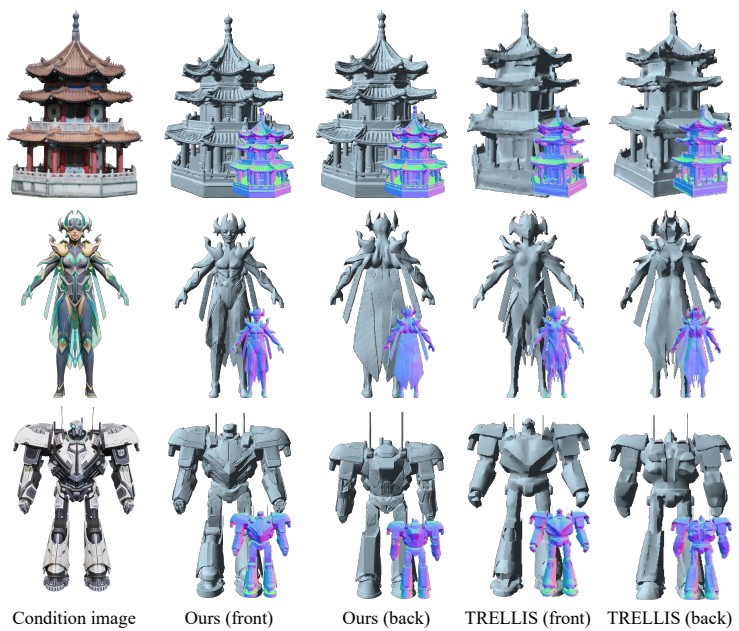

| Condition image | Ours (front) | Ours (back) | TRELLIS (front) | TRELLIS (back) |

Figure 6: **Qualitative comparison of single-image-to-3D generation.** Under the same architecture and model size [32], the generator trained with our Sparconv-VAE yields more detailed reconstructions than TRELLIS [32]. *Best viewed with zoom-in.*

Table 3: **Ablation study on Steps 3 and 4 in the preprocessing.** Chamfer Distance (CD, $\times 10^4$), Absolute Normal Consistency (ANC, $\times 10^2$), and F1 score (F1, $\times 10^2$) are reported.

| w/ Step 3 | w/ Step 4 | CD $\downarrow$ | ANC $\uparrow$ | F1 $\uparrow$ |
|:---:|:---:|:---:|:---:|:---:|
| ✓ | | 0.97 | 77.34 | 88.13 |
| | ✓ | 0.95 | 77.81 | 88.73 |
| ✓ | ✓ | 0.92 | 78.08 | 89.52 |

## 4.3 Ablation Studies

**Conversion cost.** Compared to existing remeshing methods [2, 15, 39], our Sparcubes achieves substantial speedups: at a 512-voxel resolution, conversion takes only around 15 s—half than [2, 15, 39]—and at 1024-voxel resolution, it completes in around 30 s versus around 90 s by [2, 15, 39]. Moreover, by eliminating modality conversion in our VAE design, we avoid the additional SDF resampling step, which in earlier pipelines adds roughly 20 s at 512 resolution and about 70 s at 1024 resolution [2, 15, 39]. Detailed performance comparisons can be found in the Supplementary Material.

**Training cost.** Thanks to our modality-consistent design, Sparconv-VAE converges less than two days—about four times faster than previous methods, *i.e.*, sparse voxel–based TRELLIS [32] and vecset-based approaches [2, 15] each require roughly seven days to train.

**VAE with 2D rendering supervision.** We also investigate the effect of incorporating 2D rendering losses into our VAE by using the mask, depth, and normal rendering objectives. We find that adding 2D rendering supervision yields negligible improvement for our Sparconv-VAE. This observation concurs with Dora [2], where extra 2D rendering losses were likewise deemed unnecessary for a 3D-supervised VAEs. We attribute this to the fact that sufficiently dense 2D renderings encode essentially the same information as the underlying 3D geometry—each view being a projection of the same 3D shape.

**Ablation of each in the preprocessing.** We perform an ablation study on Steps 3 and 4 to isolate the contributions of deformation optimization and rendering-based refinement:

In addition to consistent quantitative gains across all metrics, enabling both steps leads to substantial visual improvements—often beyond what the metrics alone can capture in terms of perceptual quality.

## 5 Conclusion

We introduce Sparc3D, a unified framework that tackles two longstanding bottlenecks in 3D generation pipelines: topology-preserving remeshing and modality-consistent latent encoding. At its heart, Sparcubes transforms raw, non-watertight meshes into fully watertight surfaces at high resolution—retaining fine details and small components. Building on this, Sparconv-VAE, a sparse-convolutional variational autoencoder with a self-pruning decoder, directly compresses and reconstructs our sparse representation without resorting to heavyweight attention, achieving state-of-the-art reconstruction fidelity and faster convergence. When coupled with latent diffusion (e.g., TRELLIS), Sparc3D elevates generation resolution for downstream 3D asset synthesis. Together, these contributions establish a robust, scalable foundation for high-fidelity 3D generation in both virtual (AR/VR, robotics simulation) and physical (3D printing) domains.

**Limitations.** While our Sparcubes remeshing algorithm excels at preserving fine geometry and exterior components, it shares several drawbacks common to prior methods. First, it does not retain any original texture information. Second, when applied to fully closed meshes with internal structures, hidden elements will be discarded during the remeshing process.

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
