# OpenReview forum: "Sparc3D: Sparse Representation and Construction for High-Resolution 3D Shapes Modeling"
_NeurIPS.cc/2025/Conference — NeurIPS 2025 poster_

### Official Review · Reviewer_L8c7 · 2025-06-30

**Clarity:** 3
**Significance:** 2
**Originality:** 2
**Rating:** 3
**Confidence:** 2

**Summary:**

This paper presents SparCubes, a fast and near-lossless remeshing algorithm that produces watertight 3D surfaces at high resolution in just 30 seconds, offering a 3× speedup over prior work. The authors also propose SparConv-VAE, a modality-consistent variational autoencoder with a sparse convolutional encoder and a self-pruning decoder, achieving efficient and accurate reconstruction without relying on global attention.

**Questions:**

- How does utilizing a sparse set of active voxels maintain 3D reconstruction accuracy compared to processing a dense voxel grid across the entire bounding volume?

- How can flood-fill algorithms reliably distinguish between interior and exterior regions when the target object lacks watertight closure during initial training stages?

- Could directly optimizing the geometric structure of sparse cubes instead of explicitly refining a globally accurate SDF lead to loss of fine surface details?

- What are the advantages of decoupling SDF prediction into sign classification (using BCE loss) and magnitude regression (L2 loss)? Has end-to-end SDF regression with sign-aware loss functions been explored as an alternative?

- In Table 2, many of the metrics fall short compared to previous work. The authors should provide an analysis to explain these performance gaps.

- For the visualization comparisons, the authors should include more published methods as baselines, such as XCubes.

**Ethical Concerns:**

["NO or VERY MINOR ethics concerns only"]

**Final Justification:**

Thanks very much for the rebuttal. After reading the rebuttal and other reviews, I keep my original rating. Specifically, I am not quite convinced with the answer to my last question.

**Limitations:**

Yes

**Quality:**

2

**Strengths And Weaknesses:**

- The authors propose a method to address high-resolution 3D shape modeling, which is an interesting and relevant topic. Moreover, they conduct extensive experiments to validate the effectiveness of the proposed approach. In addition, the proposed method achieves this with less computational time compared to existing methods.

- The use of sparse voxels over dense grids lacks sufficient justification, and directly optimizing sparse structures may lead to loss of fine details.

- The reliability of the flood-fill algorithm on non-watertight shapes is questionable, and the decoupled SDF prediction lacks comparison with end-to-end alternatives.

- Several metrics in Table 2 fall behind prior work without explanation, and the visualization comparisons lack key baselines such as XCubes.

- The current method struggles to preserve fine details of the target and is prone to losing small or hidden components, making it challenging to apply in practical projects.

---

> ### Author Rebuttal · Authors · 2025-07-31
>
> We would like to thank for your comments. Below, we address each of your concerns in detail.
>
> **(W1&Q1) The motivation of using sparse voxels:** The motivation for using sparse voxels is both well-founded and straightforward. Dense grids face significant limitations in terms of memory and computational efficiency. Direct optimization in dense grids becomes infeasible due to out-of-memory issues when the resolution exceeds $256^3$, making it practically impossible to operate at resolutions like $1024^3$.
>
> Our sparse voxel representation does not compromise fine details. In the ideal case, a dense grid activates voxels along the surface of the raw mesh—precisely what our encoder achieves with sparse cubes. Thanks to the high efficiency of our encoder, we can afford much higher resolutions than dense grids, resulting in superior reconstruction and generation performance.
>
> **(W2) The reliability of the flood-fill algorithm:** Apart from our flood fill approach, the previously most common method for determining inside/outside regions in non-watertight remeshing is computing the winding number. However, winding number requires the input mesh to have correctly oriented normals—an assumption that is often violated, especially in large-scale mesh datasets from the internet where normal directions are frequently incorrect or inconsistent. In contrast, our flood fill method only requires the initial seed point to be outside the surface (e.g., placed in a corner), making it a more stable and controllable solution.
>
> **(W3&Q4) The metric that fall behind:** ANC on Objeverse is one of the few metrics where our method does not consistently outperform others. However, ANC evaluates absolute normal consistency against the ground-truth normals, which are often noisy or incorrect. Our method, on the other hand, corrects the normal orientations via flood fill. Therefore, this metric may not accurately reflect the actual surface quality. Moreover, as shown in the visual results from Objaverse in Fig. 5, our method preserves significantly more geometric details than CraftMan—beyond what is indicated by the quantitative metrics.
>
>
> **(W4) Fine details and small or hidden components:**
> As highlighted in our paper, e.g., the teaser figure demonstrates rich details and hidden components, we argue that our method more effectively preserves fine details, including those in occluded regions. Furthermore, as shown in Fig. 5, our approach retains small and hidden structures without loss compared with the existing SOTA method. These components are already well captured during our multi-step process (Steps 1–3). Compared to prior methods using 2D supervision like Trellis, our approach demonstrates significantly better preservation of hidden parts, as the 2D rendering loss used provides very weak supervision for unobserved areas. Unlike many existing open-source and even commercial text-to-3D models, our generated meshes are strictly watertight, making them suitable for practical applications such as 3D printing.
>
> **(W4&Q5) Visulization of XCubes:**
> We already provided the visualization results of XCubes in the last row of Fig. 5 of the main paper.
>
> **(Q2) Flood-fill for the object w/o watertight closure:**
> We address this challenge using a coarse-to-fine multi-scale flood-fill strategy. Even when the target object is not fully watertight during early training stages, the coarsest scale provides a sufficiently smoothed and regularized representation that enables robust separation between interior and exterior regions. This coarse-level initialization ensures reliable flood-fill propagation, which is then progressively refined at finer scales.
>
> **(Q3) Refining a globally accurate SDF:**
> Our globally accurate SDF is in fact obtained through local optimization over high-resolution sparse cubes, rather than by fitting a global function directly. Therefore, it does not suffer from the loss of fine surface details.
>
> Thanks to the high resolution of our sparse representation, local SDF refinement is both efficient and precise. Moreover, the sparse cubes are tightly aligned with the raw mesh surface, effectively covering the local geometry. Optimizing these cubes is essentially equivalent to optimizing accurate local SDFs, which preserves detailed structures.
>
> **(Q4) Decoupled SDF prediction:** As with other large-scale generative models, conducting a comprehensive ablation study on each individual loss term is infeasible. Nevertheless, we argue that decoupling the sign and magnitude of the SDF is well justified for two reasons:
>
> - **Empirical success in 3D vision.** Prior work on implicit surface learning (e.g., Points2Surf [1]) has demonstrated that separating the prediction of sign (inside vs. outside) from the prediction of distance yields significantly better results than treating the SDF as a single end-to-end regression target.
> - **Faster convergence for classification vs. regression.** Statistical learning theory shows that, under low-noise (margin) conditions, classification can achieve strictly faster minimax convergence rates than regression. In particular, Audibert and Tsybakov [2] construct plug-in classifiers that attain “fast” and even “super-fast” rates—up to $O(n^{-1}$)—for the excess Bayes risk when the conditional probability function satisfies the Tsybakov margin assumption.
>
> We hope that our responses adequately address your concerns and demonstrate the soundness of our approach.
>
> **References:**
>
> [1] Erler, P., Guerrero, P., Ohrhallinger, S., Mitra, N. J., & Wimmer, M. (2020). Points2Surf: Learning implicit surfaces from point clouds. In ECCV (pp. 108–124). Springer.
>
> [2] Audibert, J.-Y., & Tsybakov, A. B. (2007). Fast learning rates for plug-in classifiers.

---

> ### Comment · Reviewer_L8c7 · 2025-08-05
> **Re Rebuttal**
>
> Thank you to the authors for their efforts in addressing my concerns. The responses have clarified several issues, including the reliability of the flood-fill algorithm, the preservation of fine details, and the visualization results. However, I am not sure if I understand the explanation of how sparse sampling can reliably capture critical information within the target. In addition, not sure about the authors' claim that existing evaluation metrics fail to accurately reflect the effectiveness of their method.

---

> > ### Author Response · Authors · 2025-08-06
> >
> > Thank you very much for your feedback and for acknowledging the clarifications we provided regarding the flood-fill algorithm, fine detail preservation, and visualization results.
> >
> > Based on our understanding, we provided additional explanations below for the points we think might still be unclear to you. We hope these clarifications address your remaining concerns. **Please feel free to reach out if you have any further questions before you submit the final rating.**
> >
> > 1. On sparse sampling and capturing critical information:
> >
> > Our sparse sampling **preserves all the information needed to reconstruct the input mesh**: we select and process sparse voxels whose eight SDF values are not all identical in sign, i.e., voxels intersected by the mesh. In other words, assuming infinite network capacity, our method would **achieve the same performance as a dense voxel representation**. For dense voxels, if all eight SDF values in a cube are either positive or negative, no mesh will be generated in that region (e.g., Marching Cubes would skip it).
> >
> > Although the theoretical performance upper bound is the same for both representations, our advantage is clear: the sequence length of our modeled representation is **significantly shorter** than that of the dense one, which is crucial for the subsequent generation.
> >
> >
> > 2. On the reliability of evaluation metrics:
> >
> > We understand the concern about questioning the accuracy of existing metrics. **Our argument is not that the metrics (ANC) themselves are inherently flawed, but that their inputs can be problematic.** Such as incorrect normal directions in ground truth: Some datasets, such as watertight spheres, contain inward-pointing normals, while the visually and physically correct interpretation should be outward. This misalignment leads to misleading normal evaluation results.
> >
> > In short, when these cases occur, a method may generate perceptually and geometrically correct results but still be penalized by the metric due to incorrect or ambiguous ground truth.

---

### Official Review · Reviewer_6geV · 2025-06-30

**Clarity:** 3
**Significance:** 3
**Originality:** 3
**Rating:** 5
**Confidence:** 5

**Summary:**

The paper proposes a new 3D representation for reconstruction and generation, named SparCubes. The representation builds upon sparse and deformable vertices and cubes. Different from the original marching cubes, the sparse vertices are optimized to move to the boundary of meshes. The SDF values and position differences are further optimized by 2D depth and mask renderings. Based on this representation, the paper presents a VAE and a subsequent diffusion model on the vae latents.

**Questions:**

- In the step 3 of SparCubes reconstruction, are the SD values on vertices also optimized along with the delta positions? Are all vertices optimized to move towards the boundary? If it is the case, does the spatial grouping of many vertices lead to unstable optimization, such as face flipping?
- In eq 5. does the delta function mean the per vertex shift? What does approximately equal mean here, does it mean that there is a constraint on the movement of the vertex positions?
- Reference [28] (TripoSR) is not a diffusion model but a LRM. In this context, it seems to make more sense to replace it with "TripoSG: High-Fidelity 3D Shape Synthesis using Large-Scale Rectified Flow Models".
- "x" in Eq 3. should be math-bolded.

**Ethical Concerns:**

["NO or VERY MINOR ethics concerns only"]

**Final Justification:**

The authors address most of my concerns in rebuttal. The paper presents a novel approach in encoding and generating 3D meshes and achieves better reconstruction quality compared to previous methods. Despite the required per-object optimization, I still find the 3D representation valuable for the community. So I will maintain my rating as Accept.

**Limitations:**

Yes.

**Quality:**

3

**Strengths And Weaknesses:**

Strengths:
- The proposed method is effective in converting meshes with open surfaces to watertight meshes, which is crucial in 3D generative modeling. The remeshing process brings little loss to model accuracy.
- By converting the meshes to the SDF-based reprensentation, it facilitates the training of the 3D VAEs, involving no modality conversion but only compressing.
- The reconstruction and generation results look superior than previous methods.

Weaknesses:
- SparCubes is implemented with custom CUDA kernels. I have concerns about the reproducibility if the code is not open sourced.
- In the SparCubes reconstruction pipeline, the step 3-4 involve per-object optimization, which seems time-consuming. I am not sure whether the claimed time (line 63) includes these steps.
- SparseFlex is not compared in the main paper, especially under quantitative evaluations.
- Some technical details are missing. Please see questions below.

---

> ### Author Rebuttal · Authors · 2025-07-31
>
> We thank Reviewer 6geV for the thoughtful and constructive feedback, and for recognizing the value of our work.
>
> Below, we provide our detailed response:
>
> **(W1) Reproducibility of custom CUDA kernels:** As researchers, we wholeheartedly agree that open-source is essential for the community. We are currently seeking internal approval and, once granted, will release the code publicly. Moreover, in the camera-ready version, we will include a detailed description of our CUDA kernels to ensure full reproducibility.
>
> **(W2) Optimization cost in remesh steps 3–4:** Our reported timing already includes the optimization overhead for steps 3–4. Thanks to the cubes’ high initial resolution—where they closely align with the target surface—each cube requires only a few updates:
> - Step 3: converges in fewer than 10 gradient updates.
> - Step 4: converges in approximately 100 gradient updates.
>
> This efficiency enables the full per-object optimization to complete in roughly 30 seconds.
>
> **(W3) Comparison to SparseFlex:** We appreciate the reviewer’s suggestion. SparseFlex is an concurrent arXiv preprint (uploaded on March 27, 2025), just prior to our NeurIPS submission deadline, so we originally placed the quantitative comparison in the supplementary material. In the camera-ready version, we will move this comparison into the main manuscript for greater visibility.
>
> Below, we report Chamfer Distance (CD) and runtime under challenging evaluation settings:
>
> |Method | CD ($\times10^4$) ↓ | Token Num ↓ | Enc Dec Time (s) ↓ |
> |:---|:---:|:---:|:---:|
> |Ours | 0.88 | 46047| 2.2 |
> |SparseFlex | 0.87 | 265601| 13.0 |
>
> As shown, our method achieves a comparable Chamfer Distance while using ~6× fewer tokens and delivering ~6× faster encoding–decoding, consistent with the qualitative results presented in the supplementary material.
>
>
> **(Q1) Details about remeshing optimzation:** We jointly optimize per-vertex SDF offsets and positional deltas, enforce sign consistency on the SDF (so that positive/negative labels never flip), and constrain each vertex’s movement to within half of the original cube edge—provably preventing any inverted faces. We will make these details explicit in the camera-ready version.
>
> **(Q2) Typo in Eq. 5:** We apologize for the typo in Eq. 5: the expression $ \delta(x) \sim \delta(x) $ was included by mistake and will be removed. We did not rely on this assumption.
>
>
> **(Q3) Typo in Reference [28] (TripoSR):** We thank the reviewer for catching this error. In the camera-ready version, we will correct the citation to:
>
> ''TripoSG: High-Fidelity 3D Shape Synthesis using Large-Scale Rectified Flow Models.''
>
>
> **(Q4) Formatting in Eq. 3:** We appreciate the reviewer’s careful attention to detail. We will boldface 'x' in Eq. 3 in the camera-ready manuscript.

---

> ### Comment · Reviewer_6geV · 2025-08-04
>
> I appreciate the response from the authors, which addresses my concerns largely. Please carefully include these quantitative comparisons in the camera-ready version. After also reading other reviews and rebuttals, I think this is a good paper to present in NeurIPs.

---

> > ### Author Response · Authors · 2025-08-06
> >
> > Dear Reviewer,
> >
> > Thank you for your update and for recognizing our work! We appreciate the time and effort you’ve dedicated to reviewing our manuscript and will revise it according to your suggestions.
> >
> > Best regards,
> >
> > Authors

---

### Official Review · Reviewer_Awed · 2025-07-01

**Clarity:** 2
**Significance:** 2
**Originality:** 3
**Rating:** 4
**Confidence:** 5

**Summary:**

This paper introduces SparC, a unified framework for high-resolution 3D shape modeling that addresses key limitations in existing 3D generation pipelines. The work consists of two main components: SparseCubes, a sparse deformable marching cubes representation that converts raw meshes into watertight surfaces at $1024^3$ resolution, and SparConv-VAE, a VAE based on sparse convolution architecture achieved SOTA performance when trained on the high-quality processed watertight meshes.

The paper's main contributions are:
- A watertight remeshing algorithm, i.e., SparCubes, that preserves fine details and small components while achieving nearly 3 times speedup over baseline methods.
- A space-conv-based VAE for SparCubes encoding.
- State-of-the-art 3D shape reconstruction quality.

**Questions:**

Please refer to major weaknesses for more details.

- Remesh algorithm comparisons and ablation study.
- The novelty of sparconv-vae is not clear to me.
- Model architecture and training details are missing.

Overall, I think this paper is not ready for publication in NeurIPS. I will consider raise my score if the authors can address my questions.

**Ethical Concerns:**

["NO or VERY MINOR ethics concerns only"]

**Final Justification:**

After careful consideration of the authors' responses, I am updating my evaluation and raising my score to borderline accept. While some areas still warrant further improvement and more detailed explanations, the authors have adequately addressed the majority of the concerns raised during the review process.

However, I strongly recommend that the authors follow through on their promises to:

- Incorporate the remeshing algorithm ablation studies into the main paper, as these studies are crucial for understanding the method's performance
- Include detailed model architecture descriptions in the main paper to enhance reproducibility and technical clarity
- Add more comprehensive remeshing visualization comparisons to better demonstrate the qualitative improvements achieved by the proposed method

These additions would significantly strengthen the paper's contribution and make it more accessible to the research community.

**Limitations:**

While the authors mention some limitations in their conclusion section (texture loss and internal structure handling), the limitations discussion is insufficient and should be expanded to address several critical aspects:

- Failure case analysis. The paper lacks discussion of failure modes. For example, what types of mesh structures cause the deformation optimization to converge poorly?
- The technical limitations of the proposed remeshing algorithm is not clear and should be analyzed.

**Quality:**

3

**Strengths And Weaknesses:**

### Strengths

- The paper is well-written and easy to follow.
- The paper proposes an engineering solution to address the high-resolution watertight remeshing problem, which is a key bottleneck in current 3D generation workflows.
- The paper achieves state-of-the-art reconstruction results both quantitatively and qualitatively across several benchmarks

### Major Weaknesses

- While this work focuses on remeshing algorithms, it only compares against Dora's remesh algorithm, which is a baseline implementation in their open-source code. The work should compare against more advanced remeshing algorithms and provide detailed analysis of advantages and disadvantages.
- SparCubes involves 4 steps, but the ablation study lacks analysis of individual components. For example, the actual contribution of deformation optimization and rendering refinement steps is unclear to reviewer.
- TRELLIS already adopts sparse convolution, but the differences and advantages of SparConv-VAE compared to TRELLIS's VAE are not clearly illustrated. It is hard to determine whether the improvements over TRELLIS mainly come from higher-quality remesh data or VAE architecture design.
- The paper lacks clear descriptions of VAE and latent flow model details, including model architecture and latent resolution. Moreover, TRELLIS adopts a two-stage training approach - it's unclear whether this work follows a similar scheme.

**Minor weaknesses**
- Section 3.3 mentions hole filling but doesn't clearly explain where this algorithm is applied or demonstrate its effectiveness.
- No analysis of SparCubes' resolution bottlenecks and failure cases.
- The contribution of each loss term during vae model training is unclear.

---

> ### Author Rebuttal · Authors · 2025-07-31
>
> We appreciate the time and effort dedicated to evaluating our work, and we have addressed each point raised with detailed clarifications
>
> **(W1) Additional Baselines:**
> We have compared our method against the state-of-the-art remeshing techniques available at the time of submission (e.g., Dora). In this rebuttal, we also include results for Mesh2SDF [1] on the most challenging cases. We would be happy to add further comparisons if the reviewers can suggest any other readily available, advanced remeshing algorithms.
>
> |Name|	CD ↓ (×10⁴)|	ANC ↑ (×10²)|	F1 ↑ (×10²)|
> |:---|:--:|:--:|:--:|
> |Dora|	1.32|	75.23|	79.10|
> |Mesh2SDF [1]|	2.10|	72.19|	76.19|
> |Ours |	0.92|	78.08|	89.52|
>
> **(W2) Ablation of each in the preprocessing:**
>
> We perform an ablation study on Steps 3 and 4 to isolate the contributions of deformation optimization and rendering-based refinement:
>
> |w/ Step 3|	w/ Step 4|	CD (×10⁴) ↓|	ANC (×10²) ↑|	F1 (×10²) ↑|
> |:--:|:--:|:--:|:--:|:--:|
> |✓|		|0.97|	77.34|	88.13|
> |	|✓	|0.95|	77.81|	88.73|
> |✓	|✓	|0.92|	78.08|	89.52|
>
> In addition to consistent quantitative gains across all metrics, enabling both steps leads to substantial visual improvements—often beyond what the metrics alone can capture in terms of perceptual quality. Due to limitations of rebuttal this year, we cannot include these visuals here; however, we will present comprehensive side-by-side comparisons in the camera-ready version.
>
>
> **(W3) Comparsion with the Trellis' VAE:**
> We would like to clarify that TRELLIS's VAE is based on sparse transformers rather than sparse convolutions. Its architecture is fully composed of Swin attention blocks, which are not scalable or trainable at high resolutions (e.g., 1024³), due to the prohibitive memory and computational cost. This makes direct comparison with our approach technically infeasible.
>
> Our VAE, in contrast, offers significant advantages: It achieves extremely high compression ratios while maintaining superior reconstruction quality and effective learning for downstream generation tasks. Specifically, it supports decoding from a latent resolution of $64^3$ to $1024^3$, significantly surpassing the $64^3→256^3$ decoding capability in TRELLIS.
>
> We did not include an explict concept comparison with the TRELLIS VAE because the two are fundamentally different in terms of representation space, training objectives, and architecture. For example:
>
> - Our VAE learns native geometric representations, whereas TRELLIS relies on lossy DINO features.
> - TRELLIS lacks sparse downsampling and upsampling modules, making it infeasible to handle high-resolution sparse inputs. Dense occupancy grids at this scale become too large to train/inference and result in out-of-memory (OOM) issues.
> - We use supervision in the 3D space while Trellis use 2D supervsion from the image space.
>
> Lastly, we would like to emphasize that our remeshing pipeline is itself a key contribution of this work, and forms an integral part of the overall design.
>
> **(W4) Detailed architecture:** The detailed design of our VAE is provided in the supplementary material. As noted in the paper, the architecture of the generative component is identical to that of TRELLIS, and we adopt the same two-stage design.
>
> **(MW1) Where to apply hole-filling:** We apply hole-filling to the outputs of the second stage—that is, the final generated results. While this does not affect the VAE evaluation metrics, it can influence the visual quality of the generated meshes.
>
> **(MW2) Resolution bottleneck and failure cases:** The resolution of SparCubes is currently limited by the nvdiffrast rendering backend. In particular, nvdiffrast imposes a fixed upper limit on the number of mesh fragments due to its internal mesh ID encoding. Therefore, we could not directly applied it to the resolution of $2048^3$.
> However, this is a constraint of the underlying renderer rather than our framework design.
>
> As for failure cases, SparCubes may struggle with meshes that contain extremely occluded internal parts. In such cases, Step 4 may not be effective due to the lack of visibility during optimization. While Steps 1–3 can still provide a coarse reconstruction of these hidden areas, the fidelity may not match that of the visible surfaces.
>
> However, compared to Trellis, SparCubes demonstrates significantly better internal structure reconstruction. This is because our Steps 1–3 are unaffected by occlusion and do not rely on 2D rendering loss for supervision. For example, the internal details of the robot in the teaser figure are well preserved in our results, whereas Trellis—being solely guided by 2D supervision—tends to lose such internal structures.
>
>
> **(MW3) Ablation of each training loss**
> We validated the effectiveness of each loss term under a small-scale dataset prior to the scaled-up training. As with many large-model frameworks, conducting a full ablation study may be impractical due to computational constraints.
>
>
> **Technical limitations**
> - We observed that, apart from the commonly known challenge of handling highly occluded internal structures—which remains difficult for most methods—our approach does not exhibit any significant additional limitations in terms of the remeshing quality compared to existing methods.
> - Similar to existing methods, our use of Marching Cubes can result in meshes with a high number of faces. In certain application scenarios, this may require additional post-processing, which is also a common open problem in the community. Nonetheless, we believe our compact latent representation could benefit downstream retopology tasks, making this a promising direction for future exploration.
>
> **Reference:**
>
> [1] Peng-Shuai Wang, Yang Liu, and Xin Tong. 2022. Dual Octree Graph Networks for Learning Adaptive Volumetric Shape Representations. ACM Trans. Graph. (SIGGRAPH) 41, 4, Article 103 (July 2022), 14 pages.

---

> > ### Comment · Reviewer_Awed · 2025-08-07
> > **Official Comment by Reviewer Awed**
> >
> > Thank you to the authors for their efforts in addressing my concerns. The clarifications have significantly improved my understanding of the work, particularly regarding the impact of different remeshing algorithm steps and their potential limitations, as well as the model's architectural design and its comparison with TRELLIS. I am strongly recommend that the authors incorporate the remeshing algorithm ablation studies and detailed model architecture descriptions into the main paper.
> >
> > Additional Questions for Further Clarification:
> > While the responses are helpful, I have two follow-up questions:
> >
> > Quantitative VAE Comparison: You mention superior compression ratios and reconstruction quality compared to TRELLIS. Could you provide specific quantitative metrics (e.g., compression ratios and reconstruction metrics) to support these claims?
> >
> > Computational Efficiency: Can you provide training time and memory consumption comparisons between your method and TRELLIS at comparable resolutions (e.g., 64³→256³)?

---

> > > ### Author Response · Authors · 2025-08-07
> > >
> > > Thank you for your continued engagement and for recommending that we integrate the remeshing ablation studies and detailed architectural descriptions into the main manuscript. Below, we address your follow-up questions in detail.
> > >
> > > 1. Quantitative VAE Comparison
> > > Both our method and TRELLIS encode a latent tensor of spatial resolution 64³ multiplied by the same sparsity ratio. The key difference lies in channel count and target mesh resolution: our model uses 16 channels to represent a 512³ mesh, whereas TRELLIS uses 8 channels for a 256³ mesh.
> > > Hence, the compression ratio—defined as the ratio of total voxels to latent scalars per voxel—is
> > >
> > > $$
> > > \frac{512^3}{64^3 \times \text{sparsity} \times 16} \text{(Ours)}
> > > =4\times
> > > \frac{256^3}{64^3 \times \text{sparsity} \times 8} \text{(Trellis)},
> > > $$
> > >
> > > demonstrating a 4× higher compression ratio than TRELLIS.
> > >
> > >  (We apologize for the earlier typo in the rebuttal referencing a 64³ → 1024³ target; the correct is 64³ → 512³ and 128³ → 1024³.)
> > >
> > > 2. Reconstruction Quality
> > > As reported in Table 2, across multiple benchmarks—including ABO, Objaverse, and In-the-Wild—our method consistently outperforms TRELLIS among all mertics, especially achieving over 30% improvement in Chamfer Distance. Fig. 5 further illustrates gains in details, confirming superior reconstruction fidelity.
> > >
> > > 3. Computational Efficiency
> > > To ensure an apples-to-apples comparison, we adapted our VAE to a two-stage downsample/upsample pipeline (256³ → 64³ → 256³), while TRELLIS retains its original single-stage (64³ → 256³) design. On a single GPU, our pipeline requires 0.4 s per training iteration and 16 GB of GPU memory, whereas TRELLIS takes 1.8 s and consumes 28 GB. Thus, our approach not only delivers higher compression and reconstruction quality but also trains 4.5× faster with substantially lower memory overhead.
> > >
> > > We hope these details address your questions. Thank you again for your valuable feedback.

---

> ### Author Response · Authors · 2025-08-06
>
> Dear Reviewer,
>
> We hope this message finds you well. As the discussion period is nearing its end with less than two days remaining, we wanted to follow up to ensure that we have addressed all your concerns satisfactorily. If there are any additional points or feedback you would like us to consider, please feel free to let us know.
>
> Your insights are invaluable to us, and we remain eager to clarify or improve any aspect of our work based on your comments.
>
> Thank you again for your time and effort in reviewing our paper.
>
> Authors

---

### Official Review · Reviewer_FtUJ · 2025-07-04

**Clarity:** 3
**Significance:** 3
**Originality:** 3
**Rating:** 5
**Confidence:** 3

**Summary:**

This work introduces a unified framework called SparC, that combines sparse deformable marching cubes representation SparseCubes with a vae-baed encoder SparConv-VAE in order to address the challenges faced by the existing two stage pipelines that can result in a major loss of details through the vae processing. The method achieves sota reconstruction results and preserves fine details of the shapes, while maintaining efficient training and inference.

**Questions:**

1. Did you evaluate your method on other datasets, in addition to the ABO, Objaverse and Wild datasets?
2. Were different seeds used in the quantitative comparison of Table 1?
3. What is the impact of Step 4 of using multiple view images on the reconstruction / generation quality? Is there a visual comparison figure to show the impact?

**Ethical Concerns:**

["NO or VERY MINOR ethics concerns only"]

**Final Justification:**

I have reviewed the paper and I think it will be a good addition to Neurips. I have also reviewed my final scores for this.

**Limitations:**

yes

**Quality:**

4

**Strengths And Weaknesses:**

Strengths:
1. The paper is well written and provides a good motivation to the proposed method.
2. The literature review also neatly categorizes the existing methods and provides the pros and cons of the related work.
3. The experiments have proper details of the metrics used and the baselines.
4. Ablation studies are included to evaluate the speedup costs and training costs.

Weaknesses:
1. The method involves an optimization process in Step 3 and it would be helpful to understand the stability of this this optimization process. Simplest could be to show it over multiple seeds.
2. A visual comparison of impact of multi-view vs single-view rendered images in step 4 would be helpful.

---

> ### Author Rebuttal · Authors · 2025-07-31
>
> We sincerely appreciate your valuable feedback and recognization of the contributions to our work. Due to the limitations of NeurIPS this year, we are unfortunately unable to upload images.
>
> **(W1) Stability of step 3**: We deeply appreciate the reviewer’s concern about the stability of our per-object optimization in Step 3. In this phase, we simultaneously refine each vertex’s signed distance function (SDF) value and its positional offset—an essential step for achieving precise watertight reconstruction. To guard against any mesh inversions or “face flips,” we enforce strict sign consistency on the SDF values, ensuring that vertices originally classified as inside or outside never change their label. Additionally, we cap each vertex’s movement to at most half of the original cube edge length, a mathematically proven safe bound that prevents any inverted topology. Thanks to our high-resolution initialization, where sparse cubes already densely cover the raw mesh surface, this carefully constrained optimization converges extremely quickly—typically < 10 steps.
>
> To confirm the robustness of this design, we ran Step 3 one hundred times with different random seeds. We observed that the Chamfer Distance (×10⁴) varied by less than 0.01, demonstrating that our method is highly insensitive to stochastic initialization and reliably reproducible. We hope this extended analysis reassures the reviewer of the stability of our approach. Per your suggestion, we will include detailed convergence discussion of the optimization in the camera-ready manuscript.
>
> **(W2&Q3) Using multiple view images in Step 4** is necessary, as single-view rendered images cannot optimize the occluded regions. As a result, details in those unseen areas will be poor if only using single-view image.
>
> **(Q1) Additional datasets**: We apologize for the unclear description of our “in-the-wild” dataset. This collection combines the most challenging examples from multiple sources—including difficult cases from GSO and scanned or artist-created samples gathered online. In the camera-ready version, we will present a detailed breakdown of the dataset’s composition and selection criteria.
>
> **(Q2) Seed among different benchmarks:** We used the same random seed across all benchmarks to eliminate potential bias. Moreover, we repeated each experiment 100 times and observed that the Chamfer Distance ($\times 10^4$) varies by less than 0.01 across runs, demonstrating that our results are robust to seed selection.
>
> Thank you again for your insightful feedback; we hope our clarifications and planned enhancements address your concerns and further strengthen our submission.

---

> ### Comment · Area_Chair_j7oY · 2025-08-04
> **Final Justification**
>
> Dear Reviewer FtUJ,
> Did the author's response resolve your questions? If necessary, please have further discussion, and please give the final justification and rating after final confirmation.
> Best regards,
> Area Chair

---

> ### Comment · Reviewer_FtUJ · 2025-08-05
>
> I would like to thank the authors for their rebuttal. I have also looked at the discussions and reviews from other reviewers which has also helped me get a better understanding of the work. I thus continue to maintain my scores.

---

> > ### Author Response · Authors · 2025-08-06
> >
> > Dear Reviewer,
> >
> > Thank you very much for your update and for your kind recognition of our work. We sincerely appreciate the time, effort, and insights you have contributed during the review process. Your suggestions are invaluable, and we will carefully revise the manuscript to address them.
> >
> > Best regards,
> >
> > Authors

---

### Decision · Program_Chairs · 2025-09-17

**Decision:**

Accept (poster)

**Comment:**

The AC and the reviewers thank the authors for their response.
This paper, through the combination of SparseCubes and SparConv-VAE, achieves for the first time a consistent input-output modality with nearly lossless 3D representation and reconstruction, enabling high-resolution 3D reconstruction and generation.

After the rebuttal, most of the reviewers’ concerns were addressed. In the end, two reviewers leaned toward direct acceptance, one leaned toward borderline acceptance, while another reviewer (L8c7) still had concerns regarding the last issue and gave a borderline rejection.

After carefully reading the paper, the reviewers’ comments, and the authors’ response, the AC believes that the contributions of this paper are significant. Although the final generation results still have some artifacts and the surfaces are not sharp enough, the effectiveness of SparseCubes as a novel 3D data processing method, along with the efficiency of SparConv-VAE, are both impressive.

The AC recommends that the authors revise the issus, add more details in the paper, and further address the reviewers’ suggestions. With these improvements, the paper can be accepted.

The AC thanks both the reviewers and the authors for their efforts, and recommends acceptance this paper.